# Circulating miRNA as a Biomarker in Oral Cancer Liquid Biopsy

**DOI:** 10.3390/biomedicines11030965

**Published:** 2023-03-21

**Authors:** Alexandra Roi, Simina Boia, Laura-Cristina Rusu, Ciprian Ioan Roi, Eugen Radu Boia, Mircea Riviș

**Affiliations:** 1Department of Oral Pathology, Multidisciplinary Center for Research, Evaluation, Diagnosis and Therapies in Oral Medicine, “Victor Babes” University of Medicine and Pharmacy, 2 Eftimie Murgu Sq., 300041 Timisoara, Romania; 2Department of Periodontology, “Victor Babes” University of Medicine and Pharmacy, 2 Eftimie Murgu Sq., 300041 Timisoara, Romania; 3Department of Anesthesiology and Oral Surgery, Multidisciplinary Center for Research, Evaluation, Diagnosis and Therapies in Oral Medicine, “Victor Babes” University of Medicine and Pharmacy, 2 Eftimie Murgu Sq., 30041 Timisoara, Romania; 4Department of Ear, Nose and Throat, “Victor Babes” University of Medicine and Pharmacy Timisoara, 2 Eftimie Murgu Sq., 300041 Timisoara, Romania

**Keywords:** liquid biopsy, oral cancer, miRNA, early diagnosis, noninvasive, saliva, biomarker

## Abstract

Oral cancer is currently challenging the healthcare system, with a high incidence among the population and a poor survival rate. One of the main focuses related to this malignancy is the urge to implement a viable approach for improving its early diagnosis. By introducing the use of liquid biopsy and the identification of potential biomarkers, aiming for a noninvasive approach, new advancements offer promising perspectives in the diagnosis of oral cancer. The present review discusses the potential of circulating miRNAs as oral cancer biomarkers identified in body fluids such as serum, plasma, and saliva samples of oral cancer patients. Existing results reveal an important implication of different miRNA expressions involved in the initiation, development, progression, and metastasis rate of oral malignancy. Liquid biomarkers can play a crucial role in the development of the concept of personalized medicine, providing a wide range of clinical applications and future targeted therapies.

## 1. Introduction

Oral cancer represents one of the leading causes of mortality worldwide, and its incidence is expected to rise in the following years, as estimated in the GLOBOCAN 2018 report [1]. One of the main concerns related to this malignancy is the late diagnosis in advanced stages, frequently implicating lymph nodes and loco-regional metastasis. This type of cancer is characterized as being a multifactorial disease, influenced by multiple genetic and exogenous factors, diet and nutrition, immune system status, and heritage [2,3,4,5].

Liquid biopsy is an appealing approach for the diagnosis of multiple pathologies, offering the possibility to introduce a noninvasive, fast, and cost-effective method that could improve treatment and prognosis. Research has focused on developing new insights based on the encountered genomic changes associated with malignancies i to identify, monitor, and evaluate the treatment response [6,7]. Body fluids such as blood, saliva, urine, and other types of samples can be used for detection, diagnosis, and further clinical investigations based on the presence of numerous potential biomarkers that can be linked to different stages of the disease. Liquid biopsies target the analysis of circulating tumor-derived material, known as “tumor circulome”, whose components can be used as direct or indirect potential biomarkers [7]. By using a liquid biopsy, the analysis can detect specific proteins, exomes, circulation tumor DNA (ctDNA), circulating tumor cells (CTCs), and circulating ribonucleic acid (RNA) [8]. Taking this into consideration, the liquid biopsy is a new step towards oncologic personalized medicine diagnoses, achieved by developing the oncology segment [9,10]. Currently, the liquid biopsy has been introduced as a complementary tool or as an alternative to the surgical biopsy procedure, overcoming the potential limits of the biopsy such as the difficult clinical accessibility in the oral posterior sites or the dissemination potential of the oral tumor [11]. The noninvasive approach, by using liquid biopsy, can provide a complex and extended molecular profile of the oral tumor environment compared to the tissue biopsy analysis [11]. Research has outlined important aspects regarding the use of the body fluids and their implications in reflecting the genetic characteristics of the tumor, their metastatic potential, and the possibility to permanently monitor the dynamic changes that occur [12,13]. Besides being an important environment in which to assess the existence of tumor biomarkers, body fluids can offer a new perspective when analyzing the tumoral environment. 

By analyzing the molecular heterogeneity of the tumor, the results can successfully complement the tumor, node, metastasis (TNM) classification by introducing the molecular profile of each individual [14]. To choose a proper treatment plan, it is necessary to understand the tumor landscape and its genetic and cellular characteristics that could influence the final outcome. Carcinogenesis is defined as a complex process that determines the initiation and progression of the tumor, based on the transcriptomic, genetic, and epigenetic alterations. The development of new approaches that target the presence of biomarkers in different types of liquid biopsies have potential as a noninvasive, fast, and repeatable method for diagnosis and surveillance [15]. The aim of the present review is to outline the potential diagnosis role of the identified altered levels of specific miRNAs encountered in liquid environments, such as in the blood, plasma, and saliva of oral cancer patients. By assessing their presence and use as biomarkers for the diagnosis of oral cancer, we can gain a new perspective for the development of further clinical applications.

### 1.1. Using Blood Samples for Liquid Biopsy

Blood samples are a reliable source for identifying various types of biomarkers that could help characterize tumor status and progression [16]. Tissue biopsy in cases of oral cancer can sometimes fail to provide complete information regarding the existence of the loco-regional metastasis of small dimensions, limiting its use as a tool for an eventual post-treatment monitorization [17]. In the case of blood samples, they can provide a reliable source of DNA and RNA for screening, diagnosis, monitorization, and prognosis evaluation. It was first identified in 1948 when analyzing blood samples from healthy individuals and outlining the existence of cell-free DNA (cfDNA) [18]. In physiological conditions, the debris and cell molecules are removed by phagocytes, which does not occur in cancer patients, resulting in a high quantity of cfDNA in their body fluids. The specific cfDNA released by the cancer cells is known as ctDNA, and many studies have discussed its potential to reflect the genetic and epigenetic alteration in cancer patients. Moreover, ctDNA can offer other cancer-related characteristics such as the turnover rate, size, stage, vascularity, and response to drug therapy [19]. The analysis of circulating tumor cells in blood samples is important for identifying tumor heterogeneity, and many studies have revealed the presence of cfDNA as a potential biomarker for the early detection and recurrence rate, with a specificity of 99% [20]. Recently, focus has been placed on the implications of extracellular vesicles, their characteristics, and miRNA charge, with a critical role in the development, progression, and metastasis rate of the malignancies [21]. During the carcinogenesis process, the tumor will permanently release specific genetic material into the bloodstream, which could represent potential biomarkers for a less-invasive approach for diagnosis and metastasis assessment [22].

### 1.2. Using Saliva Samples for Liquid Biopsy

Saliva originates from the salivary glands and has revolutionized the medical field by providing an alternative to using this type of liquid biopsy for the diagnosis of multiple diseases. In cases of oral cancer, saliva represents an appealing body fluid, considering the permanent contact and potential transfer of malignant cells and genetic material. It is a complex body fluid that is considered to contain most of the molecules that can be encountered in blood. The cancerous cells enter the salivary flow through the gingival sulcus, transcellular (active or passive transport), or paracellular routes [23], offering the opportunity to identify the presence of multiple specific biomarkers associated with the cancer initiation process, evolution, and eventual locoregional metastasis [24,25]. Nevertheless, salivary samples have numerous advantages, being a cost-effective, noninvasive approach that offers a simple collection method, making them an appealing biofluid for screening and diagnosis [26]. By analyzing saliva and potential biomarkers, they can be divided into messenger RNAs (mRNAs), proteome, microbiome, transcriptome, and metabolome [27] (Figure 1). Currently, there are only few studies that have aimed to identify oral cancer biomarkers in salivary samples, the results of which are promising.

## 2. Materials and Methods

### 2.1. Search Strategy

The search for the required articles for the present review was conducted using PubMed, Web of Science, Scopus, EMBASE, Medline, and Google Academic in September 2022. The following search terms that were used were “oral cancer”, “head and neck carcinoma”, “liquid biopsies”, “biomarkers”, “miRNA”, “circulating biomarkers”, “circulating miRNA”, “human saliva”, “blood samples”, “liquid biopsy in oral cancer”, “early diagnosis”, “genomics”, “tumor genetics”, “non-invasive diagnosis”, “oral cancer screening”, “exosomes”, “circulating tumor cells”, and all of them in various combinations. 

### 2.2. Data Selection

The articles were screened by two independent researchers to select the representative studies, the required data, and to remove any duplicates. The articles and studies that were included in this review were in vitro and human-based studies that included tissue, blood/plasma, or/and salivary samples from patients diagnosed with oral cancer and healthy controls, focusing on the presence of circulating miRNAs and their role as biomarkers. We have excluded articles that did not report the role of miRNAs as biomarkers, articles that were published in languages other than English, studies that were published before 2010, case reports, letters to editors, conference proceedings, and short communications. 

## 3. Results

Specific circulating miRNAs proved to be a promising biomarker for oral cancer, and research studies have outlined their presence in body liquids, such as in the blood and saliva of diagnosed oral cancer patients compared to healthy individuals. One of their major advantages is the fact that they are stable in these types of samples, as they are not digested by RNase, resist high-pH environments, and exist in an extracellular environment [28,29]. The results highlighted the fact that multiple miRNAs have been reported to be dysregulated in body fluids in patients diagnosed with oral cancer compared to the healthy control group, suggesting their use as a noninvasive approach for early oral cancer detection. The field of liquid biopsies provides a large amount of molecular data related to the complete description and evolution of the oral carcinogenesis.

### 3.1. miRNA as Potential Oral Cancer Biomarker in Liquid Biopsies

#### 3.1.1. Origin of Circulating miRNA

MiRNAs are represented by single-stranded, non-coding RNAs that contain approximately 19 to 25 nucleotides that are responsible for regulating gene expression by binding to the 3′ untranslated regions (UTRs) of the messenger RNA (mRNA) and influencing the inhibition, degradation, and translation of mRNA [30]. There are approximately 1500 known human miRNAs that constitute about 3% of the genes and can regulate up to 30% of the existent genes [31]. 

#### 3.1.2. Function of miRNAs

MiRNAs have an important role in controlling the cellular biological activity through their epigenetic regulation function, being directly involved in the cellular differentiation, migration, and apoptosis process [32]. Any type of dysregulation can determine the initiation and development of malignancy [33]. MiRNAs can regulate the expression of different genes that are involved in the carcinogenesis process, including genes that can be classified into tumor suppressor genes or oncogenes [34]. An important aspect is the fact that the expression of miRNA in cases of tumors and healthy tissue is different; moreover, it can be linked to a specific cancer type [35]. Multiple studies have revealed the fact that different miRNA signatures can be linked to the diagnosis, evolution, prognosis, and treatment efficiency of malignancy. Importantly, the cell-free miRNA has been noted to be stable in body fluids since it is encapsulated in different lipo–protein complexes (exosomes, microvesicles, or apoptotic bodies) (Figure 2) [36,37]. This is an important advantage, along with their high sensitivity and specificity, that has caused cell-free miRNA to become an appealing biomarker for cancer. 

Li et al. [38] conducted a study showing that, in cases of hypoxic tumor-derived cells in oral squamous cell carcinoma (OSCC) patients, miRNA expression showed increased levels of miR-148b, miR-21, and miR-205 compared to the exosomes that had a normal oxygen concentration. Their results suggest that the levels of exosomal miR-21 can be linked to cell motility and invasion action in low oxygen level conditions. Nevertheless, a connection was found between the exosomal levels of miR-21 and the lymph node metastasis rate of OSCC patients. Existing data suggest the fact that tumor-derived exosomes play a crucial role in tumor development and progression by controlling the immune system [39].

Research has outlined that miRNAs in saliva and blood are highly concentrated in exosomes, being involved in cellular translational suppression and mRNA degradation [40]. MiRNAs are secreted in body fluids that have a crucial role in cell–cell interactions [41]; moreover, the release of miRNA from the exosomes occurs due to a genetic exchange between cells [42]. Studies have revealed that the circulating miRNAs are reliable and can represent potential biomarkers for malignancies [42].

#### 3.1.3. Methods for miRNA Detection

The conventional method for quantifying the miRNA signatures is through RT-qPCR validation and next-generation sequencing [43]. Currently, this technology has been improved by developing the droplet digital polymerase chain reaction (PCR), which aimed to solve problems concerning the sensitivity of the existing method when identifying the lower transcript copy numbers, especially while investigating the exosomal miRNAs [44]. Nevertheless, as research reports have highlighted, there is a need for the development of multiplex detection methods for revealing the cancer-related miRNA signatures that can represent a more specific and faster screening approach.

### 3.2. Dysregulated miRNA Levels Associated with OSCC

During the oral carcinogenesis process, a release of tumor-related material occurs in the bloodstream and saliva of OSCC patients. By detecting the presence of upregulated or downregulated levels of circulating miRNAs, multiple studies have focused on their potential biomarker roles (Table 1).

#### 3.2.1. Serum/Plasma miRNA Expression Changes as Biomarkers in Oral Cancer

In the last few years, multiple miRNAs have been shown to be upregulated or downregulated in the serum samples of OSCC cancer patients compared to those of healthy individuals. The existence of a signature of miRNAs encountered both in tumor tissue and in the plasma of patients diagnosed with tongue cancer has been discussed. Robinowitss et al. [74] performed a comparative study that considered a total of 359 miRNAs, of which 16 of them showed different expressions in the tumor tissue compared to matched healthy tissue. Out of those 16 miRNAs, while comparing the healthy and benign tissue to the tumor tissue, seven of them were downregulated (mi-516-3p, miR-22, miR-370, miR-139-5p, miR-145-3p, miR-30c, and let-7e) and nine were upregulated (miR-27b, miR-20a, miR-200c, miR-19a, miR-28-3p, miR-151-3p, miR-20b, miR-512-3p, and has-miR-223). Importantly, all the miRNAs that were identified as being upregulated and seven of those that were downregulated were encountered in circulating exosomes. Because the circulating miRNAs in blood are linked to either protein complexes or microvesicles, they are protected from degradation, making them a reliable method for the diagnosis of oral malignancy [75].

Yan et al. [76] studied the expression of different miRNAs in the plasma of OSCC patients from China and Denmark. Their results showed that, in the case of the Danish patients, an increase in the levels of miR-148a-3p, miR-21-5p, and miR-26a-5p and decreased levels of miR-486-5p were reported. While, in case of the Chinese population, decreased levels of miR-92b-3p, miR-486-5p, and miR-375 were observed, this was not similar to the changes in miRNA that were encountered in the Danish population. An explanation for these results could be the different exposure to the risk factors and diverse genetic inheritance, considering that the plasma collection method was uniform in the two groups. Another study performed by Maclellan et al. [45] reported low levels of seven miRNAs and high levels of fifteen miRNAs in a Canadian OSCC study group compared to the controls. A study conducted by Langevin et al. [77] aimed to review the secreted exosomal miRNA of head and neck squamous cell carcinoma patients compared to normal oral epithelial cells by performing miRNA sequencing. Their results outlined a difference between the secreted exosomal miRNAs compared to the control group, and 22 miRNAs were detected only in the exosomes of oral cancer patients.

The first study that identified a modified expression of circulating miRNAs in oral cancer was conducted by Wong et al. [55]. After they performed a microarray analysis, the results revealed an overexpression of miR-184 levels in patients diagnosed with tongue cell carcinoma compared to healthy individuals. The overexpression of miR-184 was present in tissue samples as well as in the plasma levels of the patients. An important aspect that was discussed was the fact that the plasma levels of miR-184 decreased significantly after the surgical tumor resection, outlining the correlation between the tissue and plasma levels. Another research study showed that miR-184 levels are increased in the surgical margins of the tumor, rather than in other places, suggesting its potential to identify the residual disease [78,79]. Lin et al. [80] conducted a study that reported a high level of miR-24 in the plasma of oral cancer patients compared to the healthy controls. The statistical analysis showed a sensitivity of 70% and a specificity of 82% of this miRNA. This is an important aspect, considering that miR-24 was associated with the proliferation, migration, and extent of oral cancer [52]. An increased expression of miR-146a was reported by Hung et al. [71] in a study that included oral cancer patients and healthy controls, showing 79% sensitivity after differentiating the diseased patients from the controls. In this study, the levels of circulating miR-146a decreased after the tumor resection had taken place.

The implication of miRNAs in the early diagnosis of oral cancer is a current target in the research field [81]. Lu et al. [82] reported an increased level of miR-196a and miR-196b in the plasma of oral cancer patients and precancerous lesions compared to the controls. In addition, more results also revealed the fact that both miRNA levels were upregulated during the carcinogenesis process, showing an important implication of the early diagnosis [83]. The study conducted by Chang et al. [66] focused on the expression levels of miR-423-5p, miR-150-5p, and miR-222-3p in three study groups, namely, OSCC, oral leukoplakia, and healthy subjects. A decreased level of miR-222-3p in the oral leukoplakia cases and miR-150-5p and miR-423-5p were found to be upregulated in oral cancer patients. In their study, Xu et al. [84] reported upregulated serum levels of miR-483-5p in oral cancer patients compared to a healthy control group. Moreover, the upregulated level was identified in the tumoral tissue, suggesting that the dissemination in the bloodstream occurred during the carcinogenesis process. Another important finding was that the increased serum levels of miR-483-5p were correlated with the incidence of the lymph node metastasis and the TNM staging system, differentiating between the advanced and early stages.

Schneider et al. [85] aimed to compare the miRNA profile of tumoral tissue and serum samples from five patients diagnosed with OSCC. The results showed that a total of 255 miRNAs were present in the tumoral tissue samples, as well as 381 miRNAs in the serum samples. Out of these, 214 miRNAs were identified in both types of samples. Out of the 48 miRNAs that had significantly decreased levels, 30 of them were identified in the serum samples, showing their potential as oral cancer biomarkers.

Existing studies have reported that, in oral cancer, miR-24 was identified as being upregulated in the tumoral tissue and plasma of oral cancer patients [86], being responsible for uncontrolled proliferation. Plasma levels of miR-223 were also upregulated in oral cancer patients compared to controls, while its levels were downregulated in the tumoral tissue [46]. This could be explained by the fact that miR-223 is released in the bloodstream by the normal adjacent tissue as a defense mechanism to inhibit the tumoral growth [87]. Sun et al. [68] reported upregulated miR-200b-3p in the plasma of oral cancer patients compared to the control group; however, its level decreased after tumor surgical removal. The study conducted by Liu et al. [88] revealed high plasma levels of miR-187-5p identified in the oral cancer group, levels that also decreased after the tumor resection, linking the high expression of miR-187-5p to the tumor’s presence. miR-187-5p expression could be responsible for the tumoral cells’ proliferation, as well as for cell migration and the formation of cancer cell colonies. In the study performed by Lu et al. [89], the results showed that miR-31-5p was upregulated in tumoral tissue and serum samples of oral cancer patients compared to controls, with a high sensitivity in the serum samples, as well as the power of discriminating between the two groups.

Nevertheless, the presence of modified miRNA expressions encountered in tissue and serum/plasma samples from oral cancer patients could successfully introduce them in the clinical practice as circulating biomarkers linked to different stages of the disease.

#### 3.2.2. Saliva miRNAs’ Expression Changes as Biomarkers in Oral Cancer

Currently, saliva represents an appealing liquid biopsy for the diagnosis of multiple diseases. This type of biofluid is an important source of potential biomarkers for the early diagnosis, progress evaluation, and metastasis assessment of oral cancer. The “whole mouth fluid” (WMF) is represented by the total quantity of saliva that is secreted by the three major and minor salivary glands, with the contribution of the crevicular fluid. WMF is a relatively simple, noninvasive, and cost-effective sampling method that can offer important biological information regarding the health status of an individual [90,91]. The evaluation of potential salivary miRNA biomarkers has been the target of multiple research studies that have aimed to link the oral carcinogenesis process with the different expressions of the miRNAs [92].

One of the first studies was performed by Park et al. [93] and aimed to evaluate the profile of miRNAs in whole salivary and supernatant samples from oral cancer patients and healthy controls. Their results showed that the whole salivary samples contained more miRNA profiles compared to the supernatant samples, likely because they were derived from the oral desquamated epithelial cells. They identified and compared the presence of four miRNAs encountered in the salivary samples of oral cancer patients, revealing a decreased level of miR-200a and miR-125a. It is considered that the decreased level of miR-200a promotes the epithelial–mesenchymal transition of oral tumor cells [94]. The study conducted by Liu et al. [88] included salivary samples beyond the serum ones and found that miR-31 was upregulated in both types of samples. An important aspect was the fact that the miR-31 levels were significantly higher in the salivary samples compared to the plasma ones, likely due to the direct contact with the tumor cells in the oral environment. This fact could promote miR-31 as a viable salivary biomarker for oral cancer. The results of another study revealed the existence of an increased expression of miR-31 in the salivary samples of patients diagnosed with potentially malignant oral disorders compared to the control group, outlining the potential of identifying miR-31 in the early stages of oral carcinogenesis [95].

Several studies reported an upregulated expression of miR-21 and miR-184 in the saliva of oral cancer patients and patients diagnosed with potentially malignant disorders, while the presence of miR-184 was able to be used to discriminate the OSCC cases from the potentially malignant oral ones that presented dysplasia [96]. By improving the screening techniques, researchers developed the concept of miRNoma, characterizing the salivary miRNAs profiles. A total of 734 human miRNAs were analyzed by Momen-Heravi et al. [97], who identified 11 downregulated miRNAs and 2 upregulated miRNAs in the salivary samples of OSCC patients. Another research study outlined the existence of 419 dysregulated miRNAs encountered in the salivary samples of patients diagnosed with tongue squamous cell carcinoma compared to the control group [98,99].

He et al. [52], in their study, included saliva samples obtained from OSCC patients and healthy controls and concluded that miR-24-3p was upregulated, with a high sensitivity when discriminating between the OSCC group and controls, showing a promising role as a diagnostic biomarker. Another study reported decreased levels of miR-147, miR-136, miR-503, miR-323-5p, miR-632, miR-220a, miR-668, miR-877, and miR-1250 after analyzing the saliva samples of oral cancer patients compared to controls [97]. It has also been reported that miR-221 had increased levels in the salivary samples of OSCC patients, suggesting its primordial role in the carcinogenesis process [100].

#### 3.2.3. Liquid Biopsies: Expression Profiles of miRNA in Serum/Plasma and Saliva

Research has aimed to analyze specific miRNAs’ expression that can be correlated with the early diagnosis, clinical stage, progression, and metastasis rate of oral cancer patients. By introducing liquid biopsy as an alternative for the isolation of miRNAs that have a high sensitivity and specificity for a proper clinical use, the management of oral malignancy enters in a new era. There are several miRNAs that have been reported to have similar expression in the body fluids of oral cancer patients. Several studies have reported the clinical importance of the use of salivary samples, outlining their major advantage of being in permanent contact with the malignant tumor, gaining specific characteristics. Yan et al. [76] stated that the expression of miR-146a, miR-24, miR-26a, miR-31, and miR-184 was upregulated in the serum/plasma and saliva of oral cancer patients compared to the controls. The presence of the dysregulation of these miRNAs regardless of the ethnicity, environmental risk factors, and oral habits involved outlined their potential as biomarkers for early detection. Nevertheless, it has also been reported that the expression of several miRNAs can be successfully used for the evaluation of the recurrence rate in cases of oral cancer patients. The study conducted by Momen-Heravi et al. [97] showed that the elevated levels of miR-27b were identified in the salivary samples of OSCC patients, while its levels in the plasma of these patients were reduced. An analysis of miR-148a showed an increase in the plasma level of oral cancer patients compared to the levels present in salivary samples [41]; however, there are multiple studies that have focused on the presence of miR-21 which report elevated levels in serum/plasma and salivary samples.

## 4. Discussion

Oral cancer is challenging the healthcare system, being characterized by an aggressive evolution, high metastasis rate, and late diagnosis that influence the overall survival rate of the patient. It is described as a multifactorial disease that gathers the action of multiple genetic and epigenetic factors, implying an improvement in the early diagnosis of this malignancy. There has been significant interest in developing a biomarker concept by introducing the use of liquid biopsies, focusing on serum, plasma, and salivary samples. Liquid biopsy can play a crucial role in the screening, diagnosis, and further surveillance steps by highlighting the existence of circulating biomarkers. Nevertheless, as research has shown, there is high variability between patients due to the implication of various risk factors and variables in the pathogenesis of cancer, and more studies are required to standardize the use of liquid biomarkers [101]. An ideal biomarker should be obtained through a noninvasive approach and analyzed using simple and cost-effective methods, as it would therefore be stable in clinical samples; moreover, all these characteristics are fulfilled by miRNAs. The specificity characteristic of circulating miRNAs in the diagnosis of cancer is currently a debate, and it has been suggested that a panel of multiple miRNAs could provide a higher specificity related to a certain malignancy.

The use of miRNAs as potential biomarkers in liquid biopsies has been employed for a clinical diagnosis and has developed specific miRNA profiles that can be linked to various malignancies. Studies have suggested that a panel of more miRNA expressions could be the key for the diagnosis of different cancer types, such as lung, breast, or thyroid cancer [102]. Boeri et al. [103] identified that the dysregulation of 24 miRNAs could have a diagnostic and prognostic value for lung cancer. Zhang et al. [104] identified miR-21 as being overexpressed in the serum samples of lung cancer patients, modulating the tumorigenesis process. Another study suggested focusing on a panel of four miRNAs identified in serum samples (miR-126, miR-128, miR17-92, miR-375) that showed a high sensitivity, differentiating lung cancer patients from healthy individuals [105]. The results of a study that aimed to identify circulating miRNAs in the plasma of patients diagnosed with colorectal cancer revealed the existence of upregulated levels of miR-21, miR-155, miR-203, miR-145, and miR-345 with the potential to discriminate the diseased patients from healthy controls [106]. Circulating miRNAs were the subject of a study regarding the diagnosis of prostate cancer using serum samples. The results show the presence of upregulated levels of miR-141-3p and miR-375-3p in the serum samples of prostate cancer patients compared to the patients found in remission [107].

It has been reported that multiple miRNA expressions can be altered and have been identified as being associated with the debut, progression, and metastasis potential of an oral tumor. The evolution of the current technologies targets the early diagnosis by introducing a simple and effective analysis that quantifies the presence of miRNAs. Rapado-Gonzales et al. [108] evaluated the diagnosis accuracy of the dysregulated miRNAs encountered in the liquid samples from OSCC patients and healthy controls, concluding that salivary and blood miRNAs have a high accuracy when differentiating between the two groups. Moreover, their analysis suggests the fact that the use of miRNAs for a positive oral cancer diagnosis from salivary and blood samples provided an accuracy rating of up to 84%, confirming their potential role as biomarkers. Another analysis performed by Liu et al. [109] aimed to evaluate the reported sensitivity and specificity of blood and salivary miRNA levels as potential oral cancer biomarkers. They concluded that blood miRNAs showed a positive diagnosis probability of 81% and that the use of salivary miRNAs could improve the positive diagnosis by up to 80%. Moreover, another observation was the fact that the salivary miRNAs showed a lower sensitivity but a higher specificity than the blood miRNA levels.

The use of miRNAs as potential diagnostic biomarkers is based on the ability of miRNA to pair with specific mRNAs and further target more than 60% of protein encoding-genes [110], suggesting that the dysregulation of the miRNA can contribute to the development of multiple diseases, including oral cancer [111,112,113].

An important aspect that has been discussed in multiple studies was the involvement and the presence of the Human Papilloma Virus (HPV) DNA in the serum, plasma, and salivary samples of oral cancer patients. The presence of circulating HPV DNA (cfHPV DNA) and circulating tumor HPV DNA (ctHPV DNA) has been proved to be a useful indicator of disease status and disease progression [114]. Mazurek et al. [115] showed that HPV16-positive patients were diagnosed with advanced node involvement (N2 and N3), a fact that was also identified by Veyer et al. [116], revealing a positive correlation between the plasma ctHPV16 DNA and the TNM status of the patients, highlighting higher T and N in cases with increased ctHPV16 DNA load. Summerer et al. [117] aimed to evaluate the plasma levels of head and neck squamous cell carcinoma patients and detected a set of differently expressed miRNA levels (miR-191-5p, miR-142-3p, miR- 21-5p, miR-186-5p, miR-590-5p, miR-28-3p, and miR-197-3p) that could differentiate the diseased groups from the healthy controls. Nevertheless, in their study that aimed to use plasma as liquid biopsy, no significant correlation was encountered between the HPV status and the identified circulating miRNAs.

The tumor microenvironment proved to have an important implication in the development and evolution of oral cancer. The oral environment related to oral cancer has two components: the cellular components (fibroblasts, adipocytes, neurons, endocrine cells, and cells belonging to the immune system) and the extracellular matrix [118]. Studies have shown that it can be involved in the progression of the malignancy through alterations that change the macrophages’ behavior and determine a hypoxic environment. It was reported that miR-31 was also directly involved in inducing hypoxia in the microenvironment [119]. There have been discussions regarding the implications of miRNAs in the invasion and migration process of oral cancer cells by targeting specific genes of the tumor microenvironment, such as miR-148a, which targets the WNT10B gene and is responsible for an uncontrolled migration of the malignant cells [120]. Angiogenesis and lymph angiogenesis seem to also be influenced by the action of miRNAs, and the downregulation of miR-126 determines the increase in the gene expression responsible for the vascular endothelial factor A (VEGF-A) and oral cancer-associated angiogenesis and lymph angiogenesis [121]. The downregulation of miR-300 triggers the production of vascular endothelial growth factor C (VEGF-C) that is responsible for the development of lymphatic vessels and a high metastasis rate [122]. These results show that the influence of dysregulated miRNA levels upon the tumor microenvironment contributes to the development and migration of the malignant cells. Nevertheless, the altered expressions of different miRNAs have been also linked to several other malignancies such as colon cancer [123,124], lung cancer [125], ovarian and breast cancer [126,127], oral cancer, and the evolution of carcinogenesis [128].

The various expression of miRNAs in malignant and healthy oral cells and their presence in liquid biopsies such as saliva, serum, and plasma transforms the presence of miRNA into a desirable oral cancer biomarker [129,130]. The function of circulating miRNAs has been widely discussed and recent research has revealed their involvement in cell–cell communication [131], influencing the proliferation rate and metastasis and apoptosis processes [132]. Li et al. [38] showed that miR-21 has an important influence upon the migration and invasion potential of OSCC in both types of liquid samples.

Using serum in a sampling method offers accessibility to a wide range of biomarkers; moreover, studies have showed that different miRNA dysregulations such as miR-9, miR-21, and miR-483 are present in the serum of oral cancer patients and could successfully be correlated with the prognoses of those patients [84,133,134]. Similar results were reported while using saliva samples, offering important information related to the extent of the oral malignancy. A small amount of saliva can facilitate a noninvasive approach that targets early diagnosis, exhibiting changes in circulating miRNA levels. An example would be the upregulated levels of miR-31 identified in the saliva and plasma samples of oral cancer patients, as well as in samples from patients diagnosed with oral verrucous leukoplakia, compared to healthy controls. An important aspect is the fact that the higher levels of miR-31 were identified in cases of oral cancer patients. While analyzing the salivary samples, the levels of miR-31 suffered changes after the tumor resection, outlining its tumor origin [135]. Considering that the plasma and salivary levels of miR-31 were similar in early and advance stages, this suggests the opportunity of using saliva samples for the diagnosis of oral cancer [135]. Similar studies that discuss the use of salivary samples have revealed the potential of using circulating miRNA for monitoring the progression of premalignant lesions into malignant ones, showing its use as a marker for the malignant transformation [136]. Researchers have discussed the fact that 40,8% of leukoplakia cases evolve towards malignancy, a fact that reinforces the need for a potent biomarker that can assess this aspect and monitor the risk [137]. Therefore, miR-21, miR-181b, and miR-345 could act as potential biomarkers regarding this matter as their dysregulation is responsible for this pathway.

The potential role of miRNAs to predict the lymph node metastasis and the aggressiveness of tumors has been reported [138]. Chang et al. [87] showed the potential of plasma miRNA in the early diagnosis of oral cancer (miR-150-5p, miR-222-3p, and miR-423-5p), distinguishing diseased subjects from healthy ones. Moreover, the plasma levels of miR-222-3p and miR-423-5p were linked to the occurrence of tumor progression and lymph node metastasis. Moreover, Xu et al. [86] concluded that the increased serum levels of miR-483-5p in OSCC patients could be correlated with the tumor stage and metastasis, suggesting its role as diagnosis and prognostic serum biomarker. In a study conducted by Sun et al. [59], increased miR-200b-3p plasma levels were associated with a poor prognosis. Increases in miR-626 and miR-5100 levels and poor prognosis were also identified by Shi et al. [139]. The research performed by Karimi et al. [140] outlined the potential of miRNA-21 to be associated with a tumor’s resistance to chemotherapy.

Several studies have discussed the fact that a single biomarker detection is not appropriate for a diagnosis due to the high probability of false-positive or false-negative outcomes, suggesting the use of multiple biomarkers to be more effective [141]. Researchers have suggested using a panel of biomarkers that could have an accurate role in the diagnosis of oral cancer, and existent results indicate that the identification of miR-24, miR-31, miR-21, and miR-155 has the ability to discriminate between healthy individuals and oral cancer patients [142].

The discovery and use of miRNA represents a landmark for further research to improve the early diagnosis and evaluate the prognosis of diseases. It seems that there are several miRNAs that have altered levels and can be directly linked to the initiation of the oral carcinogenesis. The potential of miRNAs is not limited to the diagnosis steps, and it has been suggested that, by modulating their levels, they could be the target of the therapy. However, there are challenges related to the use of miRNAs as oral cancer biomarkers that need to be addressed, firstly by establishing a protocol for the body fluid’s collection and analysis.

## 5. Future Perspectives and Challenges

Recently, there have been significant advancements in the liquid biopsy field and its role in cancer diagnosis and prognosis evaluation. Although the technologies that allow the assessment of miRNAs have rapidly developed, they still must face challenges regarding the sensitivity and specificity of the methods for an appropriate clinical use. An important aspect to focus on is the potential of the liquid biopsies to target and separate the cancer-associated potential biomarkers from those that are released by the normal tissue. Reports have revealed the presence of the dysregulation in case of different miRNAs in serum, plasma, and saliva samples; however, their quantification and diagnosis value are still to be completely evaluated. If, at the beginning, the blood-based liquid biopsies were a common target for the analysis, in cases of oral cancer, salivary samples offer a wide perspective of an individualized assessment. Considering that, in cases of saliva, oral cancer can directly shed tumor cells, tumor DNA, and circulating miRNAs, this type of biopsy is of a great interest. As for future approaches, by gaining sensibility and specificity, and by becoming a reliable method, liquid biopsies can be integrated for a diagnosis purpose. To make liquid biopsies more efficient and the use of biomarkers a feasible option, the standardization of their use remains an obstacle. MiRNAs play an important role in the oral carcinogenesis process; moreover, their dysregulated levels have the potential to discriminate between healthy and oral cancer patients. As for further studies, focus should be placed on identifying if a single or multiple associated miRNA signatures represent a reliable source of biomarkers, minimizing the influence of individual variables. Hence, to assess the uniform distribution of the expression profiles of different miRNAs and validate them, more studies need to be performed, and a common protocol for the collection, preparation, and analysis of miRNAs during liquid biopsy needs to be implemented to become a viable option.

## 6. Conclusions

Existing scientific evidence supports the potential of the circulation of miRNAs for the diagnosis and prognosis of oral malignancy. By using liquid biopsies as a noninvasive approach, analyses of miRNA profiles in blood and saliva samples may overcome some of the limitations of a conventional biopsy. Moreover, miRNAs have proved their potential as biomarkers, offering a wide perspective in describing a personalized oral cancer landscape. The analysis of the biological fluids can become a promising early diagnosis tool, and, with further investigations and improvements of the techniques, the results are promising. In the present review, we focused on the various expressions of circulating miRNAs that have the potential to be used as oral cancer biomarkers and outlined the influence of their modified levels on oral carcinogenesis and the existing limitations related to the clinical use of liquid biopsies.

## Figures and Tables

**Figure 1 biomedicines-11-00965-f001:**
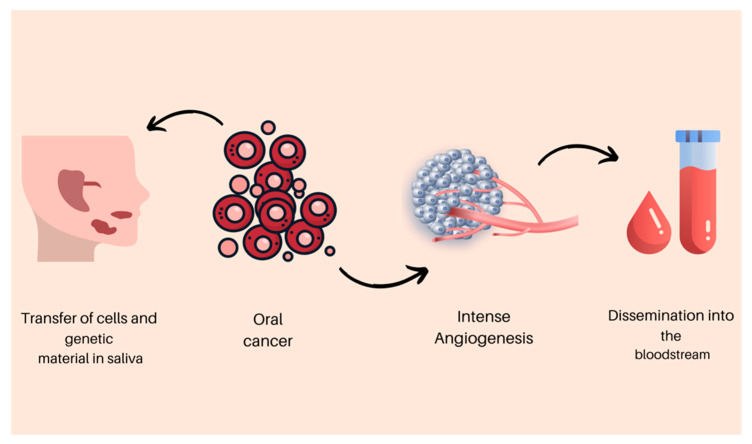
The carcinogenesis process and malignant cell dissemination.

**Figure 2 biomedicines-11-00965-f002:**
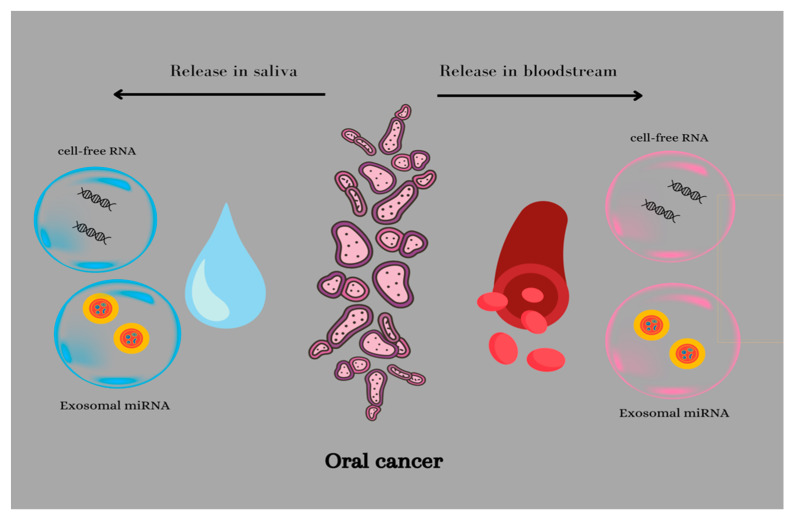
Release miRNA mechanism of oral cancer cells.

**Table 1 biomedicines-11-00965-t001:** Dysregulated miRNAs and their function in oral cancer carcinogenesis.

miRNA	Type of Sample	Level of Expression	Function	Biomarker Role	Ref.
miR-223	Serum	Low	Tumor suppressor	Diagnostic and prognostic	[45]
	Plasma	High	Inhibits proliferation and apoptosis		[46]
miR-16-1	Saliva	Low	Tumor suppressor downregulating oncogenes	Diagnostic	[47]
miR-338	Serum	Low	Tumor suppressor Inhibits proliferation and metastasis of OSCC	Diagnostic and prognostic	[48]
miR-21	Saliva Plasma	High	Role in proliferation, dissemination, and metastasis occurrence	Diagnostic and prognostic	[49,50,51]
miR-24	Plasma Saliva	High High	Proliferation antiapoptotic chemotherapy resistance	Diagnostic	[52,53,54]
miR-184	Plasma	High High	Proliferation and apoptosis	Diagnostic	[55]
miR-320	Serum	High	Migration and invasion	Metastatic and progression assessment	[56]
miR-486-5p	Serum Plasma	High Low	Cell proliferation and migration apoptosis	Diagnostic	[57]
miR-181	Plasma	High	Cell migration and invasion	Metastatic and survival rate	[58]
miR-194-5p	Plasma	High	Cell proliferation	Diagnostic and survival rate	[59,60]
miR-214-3p	Plasma	High	Invasion and migration	Diagnostic and prognostic	[61]
miR-372	Plasma	High	Metastasis and lymphovascular invasion	Metastatic and prognostic	[62]
miR-31	Saliva Serum	High	Proliferation, differentiation, migration, and invasion rate	Diagnostic and prognostic	[63,64]
miR-221	Saliva	High	Proliferation, invasions, and metastasis	Diagnostic	[65,66,67]
miR-200	Saliva	Low	Differentiation, migration, and metastasis	Diagnostic and prognostic	[68]
miR-375	Saliva	Low	Proliferation, invasions, and metastasis	Diagnostic	[69]
miR-145	Saliva	High	Cell growth	Diagnostic	[70]
miR-146a	Saliva Serum	High	Increases tumorigenesis and metastasis	Diagnostic and prognostic	[71]
miR-139-5p	Saliva	Low	Cell proliferation	Diagnostic	[72]
miR-27b	Saliva	High	Cell migration and invasion	Diagnostic	[73]

## Data Availability

Not applicable.

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
