# Peer review of "Circulating miRNA as a Biomarker in Oral Cancer Liquid Biopsy"

_biomedicines, 2023, doi:10.3390/biomedicines11030965_

Round 1
Reviewer 1 Report
This review summarizes the knowledge about the different microRNAs identified in blood and saliva samples for oral cancer diagnosis. The findings seem interesting. Whereas, there are concerns needing to be addressed.
1- Since this work investigates blood and salivary miRNAs, the title of the manuscript should be modified due to salivary microRNAs are directly released by oral cancer cells into saliva.
2-In discussion section, authors should be included the findings about the diagnostic potential of blood and salivary miRNAs as diagnostic biomarkers in OSCC (PMID: PMID: 31756680, PMID: 33516682).
5-Abbreviations are not uniform along the manuscript. Please check all abbreviations.
Line 18: cell-free DNA
Line 83: cell-free DNA (cfDNA)
Line 89: free circulating DNA (cfDNA)
Line 170: OSCC
Line 254: oral squamous cell carcinoma
Line 262: TNM (Tumor-Node-Metastasis)
Line 265: oral squamous cell carcinoma
Line 328: oral squamous cell carcinoma
Line 330: oral squamous cell carcinoma
Line 335: oral squamous cell carcinoma
Line 354: oral squamous cell carcinoma
Author Response
REVIEWER 1
Thank you for taking the time to revise our manuscript and for the suggestions given in order to improve our work. We have answered made several changes, as follows:
- Since this work investigates blood and salivary miRNAs, the title of the manuscript should be modified due to salivary microRNAs are directly released by oral cancer cells into saliva.
A: We have included the term ”circulating” in the title as referring to the existence of miRNAs in liquid environments (saliva included). Nevertheless, research has stated the fact that the salivary circulating miRNA can be the result of a direct release from the oral cancer cell in the saliva, as well as through a passive transport. (We have mentioned these mechanisms. (Lines 105-111)
- In discussion section, authors should be included the findings about the diagnostic potential of blood and salivary miRNAs as diagnostic biomarkers in OSCC (PMID: PMID: 31756680, PMID: 33516682).
A: Thank you for the suggestions, we have improved our discussion section by including more information related to the potential of blood and salivary miRNAs as diagnostic biomarkers (lines 396-408).
3 -Abbreviations are not uniform along the manuscript. Please check all abbreviations.
Line 18: cell-free DNA
Line 83: cell-free DNA (cfDNA)
Line 89: free circulating DNA (cfDNA)
Line 170: OSCC
Line 254: oral squamous cell carcinoma
Line 262: TNM (Tumor-Node-Metastasis)
Line 265: oral squamous cell carcinoma
Line 328: oral squamous cell carcinoma
Line 330: oral squamous cell carcinoma
Line 335: oral squamous cell carcinoma
Line 354: oral squamous cell carcinoma
A: We have checked the abbreviations and uniformized them along the manuscript.
Thank you once again for your time and advices! Best regards!
Reviewer 2 Report
The article “Circulating miRNA as a biomarker in oral cancer liquid biopsy” is very interesting and I have some minor comments that I believe could improve the manuscript:
- Objective: What is the concrete objective of the study?
- The reference of this sentence is not very adequate: “Researchers discuss the fact that 37.5% of leukoplakia cases evolve towards malignancy, fact that urges the need for a potent biomarker that could assess this aspect and monitor the risk” [131]. Kramer, I.R.; Lucas, R.B.; Pindborg, J.J.; Sobin, L.H. Definition of leukoplakia and related lesions: An aid to studies on oral precancer. Oral Surg. Oral Med. Oral Pathol. 1978, 46, 518–539.
- References 38 and 126 are repeated
Thank you
Author Response
REVIEWER 2
Thank you for taking your time to revise our manuscript and for the suggestions made in order to improve our manuscript:
- Objective: What is the concrete objective of the study?
A: We have reformulated the objective of the study in order to be more concrete (line 72-75)
- The reference of this sentence is not very adequate: “Researchers discuss the fact that 37.5% of leukoplakia cases evolve towards malignancy, fact that urges the need for a potent biomarker that could assess this aspect and monitor the risk” [131]. Kramer, I.R.; Lucas, R.B.; Pindborg, J.J.; Sobin, L.H. Definition of leukoplakia and related lesions: An aid to studies on oral precancer. Oral Surg. Oral Med. Oral Pathol. 1978, 46, 518–539.
A: We have modified the provided information and added a proper reference for it (lines 495-497).
- References 38 and 126 are repeated
A: We have corrected in the manuscript and removed reference 126.
Thank you once again for your suggestions! Best regards!
Reviewer 3 Report
In this manuscript, the authors discuss the potential of circulating miRNAs as oral cancer biomarkers. However, it must be said that the manuscript is totally unsuitable as a review article for the following reasons.
1. Results using serum, plasma, or saliva samples were not presented in 25 of the 30 publications in Table 1.
2. Citing review articles in reviews should be avoided as much as possible. Too many reviews cited.
3. As a minor point, there are many mistakes in the title of the papers in the reference list and same papers are cited redundantly.
Author Response
REVIEWER 3
Thank you for taking you time in revising our manuscript and the suggestions given, we have made several modifications as follows:
- Results using serum, plasma, or saliva samples were not presented in 25 of the 30 publications in Table 1.
A: We have revised the included studies that were presented in Table 1.
- Citing review articles in reviews should be avoided as much as possible. Too many reviews cited.
A: We would like to mention the fact that the data provided by the included reviews was used for outlining general information regarding the subject of out manuscript. Review articles were not presented or cited as clinical studies or included in the table as primary research.
- As a minor point, there are many mistakes in the title of the papers in the reference list and same papers are cited redundantly.
A: We have corrected where there were mistakes in the title of papers and based on other received suggestions, we have included more references.
Thank you once again for your revision and comments! Best regards!
Reviewer 4 Report
The authors described "Circulating miRNA as a biomarker in oral cancer liquid biopsy" in a review style. This topic should be attractive for potential readers. The authors outlined the potential implications of the presence of miRNAs and their altered levels that can be identified in liquid biopsies such as blood and saliva and the contents was good.
I have one recommendation. They mentioned that the liquid biopsy has been introduced as a complementary tool or an alternative to the surgical biopsy procedure, overcoming the potential limits of the biopsy such as the difficult clinical accessibility in the posterior sites, or the dissemination potential of the tumor [11]. However, references related to miRNA of other carcinomas are extremely limited. They should add references of other cancer researches describing circulating miRNA.
Author Response
REVIEWER 4
Thank you for taking your time to revise our manuscript and providing suggestions in order to improve its quality. We have made several changes, as follows:
The authors described "Circulating miRNA as a biomarker in oral cancer liquid biopsy" in a review style. This topic should be attractive for potential readers. The authors outlined the potential implications of the presence of miRNAs and their altered levels that can be identified in liquid biopsies such as blood and saliva and the contents was good.
I have one recommendation. They mentioned that the liquid biopsy has been introduced as a complementary tool or an alternative to the surgical biopsy procedure, overcoming the potential limits of the biopsy such as the difficult clinical accessibility in the posterior sites, or the dissemination potential of the tumor [11]. However, references related to miRNA of other carcinomas are extremely limited. They should add references of other cancer researches describing circulating miRNA.
A: Thank you for your suggestion. We have included more information in the discussion section regarding the use of circulating miRNAs and their role in the diagnosis of other cancers (lines 392-415). Nevertheless, we have as well included more data encountered in the studies related to the role of blood and salivary miRNAs as potential biomarkers in oral cancer (lines 396-408).
Thank you once again for revising our manuscript. Best regards!